# Impact of active case finding for tuberculosis with mass chest X-ray screening in Glasgow, Scotland, 1950–1963: An epidemiological analysis of historical data

**Peter MacPherson**[1,2,3]*, **Helen R. Stagg**[4], **Alvaro Schwalb**[4,5], **Hazel Henderson**[3],
**Alice E. Taylor**[3], **Rachael M. Burke**[2], **Hannah M. Rickman**[2,6], **Cecily Miller**[7],
**Rein M. G. J. Houben**[4], **Peter J. Dodd**[8⦿], **Elizabeth L. Corbett**[2⦿]

1 School of Health & Wellbeing, University of Glasgow, Glasgow, United Kingdom, 2 Clinical Research
Department, London School of Hygiene & Tropical Medicine, London, United Kingdom, 3 Respiratory and
Bacterial Infections Group, Public Health Scotland, Edinburgh, United Kingdom, 4 Department of Infectious
Disease Epidemiology, London School of Hygiene & Tropical Medicine, London, United Kingdom, 5 Instituto
de Medicina Tropical Alexander von Humboldt, Universidad Peruana Cayetano Heredia, San Martín de
Porres, Peru, 6 Public Health Group, Malawi-Liverpool-Wellcome Programme, Blantyre, Malawi, 7 World
Health Organization, Global Tuberculosis Programme, Geneva, Switzerland, 8 School of Health and Related
Research, University of Sheffield, Sheffield, United Kingdom

⦿ These authors contributed equally to this work.
* Peter.MacPherson@glasgow.ac.uk

## Abstract

### Background

Community active case finding (ACF) for tuberculosis was widely implemented in Europe
and North America between 1940 and 1970, when incidence was comparable to many pres-
ent-day high-burden countries. Using an interrupted time series analysis, we analysed the
effect of the 1957 Glasgow mass chest X-ray campaign to inform contemporary approaches
to screening.

### Methods and findings

Case notifications for 1950 to 1963 were extracted from public health records and linked to
demographic data. We fitted Bayesian multilevel regression models to estimate annual rela-
tive case notification rates (CNRs) during and after a mass screening intervention imple-
mented over 5 weeks in 1957 compared to the counterfactual scenario where the
intervention had not occurred. We additionally estimated case detection ratios and inci-
dence. From 11 March 1957 to 12 April 1957, 714,915 people (622,349 of 819,301 [76.0%]
resident adults ≥15 years) were screened with miniature chest X-ray; 2,369 (0.4%) were
diagnosed with tuberculosis. Pre-intervention (1950 to 1956), pulmonary CNRs were declin-
ing at 2.3% per year from a CNR of 222/100,000 in 1950. With the intervention in 1957,
there was a doubling in the pulmonary CNR (RR: 1.95, 95% uncertainty interval [UI] [1.81,
2.11]) and 35% decline in the year after (RR: 0.65, 95% UI [0.59, 0.71]). Post-intervention
(1958 to 1963) annual rates of decline (5.4% per year) were greater (RR: 0.77, 95% UI

Medicine Editorial Board, UNITED STATES OF
AMERICA

**Data Availability Statement:** Data and code to reproduce analysis are available at https://github.com/petermacp/glasgow-cxr.

**Funding:** AS and RMGJH were supported by the US National Institutes of Health [grant number R-202309–71190]. HMR was funded by Wellcome [grant number: 225482/Z/22/Z]. The funders had no role in study design, data collection and analysis, decision to publish, or preparation of the manuscript.

**Competing interests:** PM is an PLOS Medicine Academic Editor.

**Abbreviations:** ACF, active case finding; BCG, Bacillus Calmette–Guerin; CDR, case detection ratio; CNR, case notification rate; UI, uncertainty interval.

[0.69, 0.85]), and there were an estimated 4,599 (95% UI [3,641, 5,683]) pulmonary case notifications averted due to the intervention. Effects were consistent across all city wards and notifications declined in young children (0 to 5 years) with the intervention. Limitations include the lack of data in historical reports on microbiological testing for tuberculosis, and uncertainty in contributory effects of other contemporaneous interventions including slum clearances, introduction of BCG vaccination programmes, and the ending of postwar food rationing.

## Conclusions

A single, rapid round of mass screening with chest X-ray (probably the largest ever conducted) likely resulted in a major and sustained reduction in tuberculosis case notifications. Synthesis of evidence from other historical tuberculosis screening programmes is needed to confirm findings from Glasgow and to provide insights into ongoing efforts to successfully implement ACF interventions in today's high tuberculosis burden countries and with new screening tools and technologies.

## Author summary

### Why was this study done?

- Tuberculosis screening is conditionally recommended by the World Health Organization for populations with a high prevalence of disease or other structural risk factors.

- There is considerable uncertainty over the optimal approaches and population impact of tuberculosis screening.

- Between 1930 and 1970, mass screening for tuberculosis was widely undertaken, in Europe and North America, but there has been little attempt to understand what effect these programmes had on the trajectory of tuberculosis epidemics.

### What did the researchers do and find?

- Over a 5-week period, in 1957, the city of Glasgow, Scotland implemented a tuberculosis screening programme comprising mass miniature X-ray of around 715,000 people supported by community mobilisation.

- Tuberculosis notification data and population demographics were extracted from city Medical Officer of Health reports between 1950 and 1963, and multilevel interrupted time series regression models were constructed to investigate the effect of the mass screening campaign on tuberculosis notifications, compared to the counterfactual scenario where the intervention had not occurred.

- Before the mass screening intervention (1950 to 1956), tuberculosis notification rates were declining at 2.3% per year, and rates doubled in the year of the intervention (1957). Post-intervention, tuberculosis notification rates declined at 5.4% per year, and there were an estimated 4,599 pulmonary notifications averted.

- Intervention effects were consistent across all 37 city wards, but showed differing effects by age group and sex.

## What do these findings mean?

- A single, rapid, and high coverage round of mass tuberculosis screening, supported by intensive community mobilisation, likely had a major impact on changing the tuberculosis epidemiology trajectory in Glasgow.

- Greater understanding of how improved housing, social conditions, and tuberculosis care and prevention contributed to this screening effect is needed.

- Synthesis of evidence from other historical tuberculosis screening programmes is needed to confirm findings from Glasgow, and to support efforts to successfully implement active case finding (ACF) interventions in today's high tuberculosis burden countries and with new screening tools and technologies.

## Introduction

Tuberculosis has been a scourge of humankind for millennia [1]. From the late 18th to the early 20th centuries, improvements in housing, nutrition, and air quality [2]—as well as progressive improvements in tuberculosis care, treatment, and prevention [3]—yielded substantial reductions in incidence and mortality in countries such as the United Kingdom [4].

By the 1940s, it was recognised that additional measures would be required to stem the source of *Mycobacterium tuberculosis* infections and prevent tuberculosis disease [5]. In line with other European and American cities in the 1940 to 1960s [6], Scotland implemented active case finding (ACF) in communities through mass chest X-ray screening campaigns [7]. To our knowledge, the largest single site tuberculosis ACF intervention ever implemented globally was in Glasgow, Scotland, in 1957 [6]. By the end of the Second World War, Glasgow had extremely high levels of social deprivation and nearly one out of every thousand people resident in Glasgow died of pulmonary tuberculosis annually [8]. Whereas incidence of tuberculosis and mortality had declined substantially in other major UK cities, progress in Glasgow had lagged considerably behind [9].

In 1974, the World Health Organization (WHO) recommended "the policy of indiscriminate tuberculosis case finding by mobile mass radiography should now be abandoned" due to the declining diagnostic yield from screening programmes [10]. Yet, currently many countries in Africa and Asia have tuberculosis epidemiological indicators similar to those of Europe and North America in the 1940s and 1950s [11]. This suggests that the experiences of ACF in cities like Glasgow are highly informative for the modern day. Indeed, the WHO made a conditional recommendation in favour of ACF for communities where the prevalence of undiagnosed pulmonary tuberculosis was >1,000 per 100,000 in 2013 [12], lowering this threshold to a prevalence of >500 per 100,000 in 2021 [13].

Action is needed to implement high-impact interventions that can change the trajectory of tuberculosis epidemics in high-burden nations. Despite renewal of interest in ACF [14]—and more recent trial evidence [15,16]—there remains considerable uncertainty over how and where ACF should be implemented, and what effects can be expected to be achieved [17].

Using comprehensive historical records, we therefore set out to estimate the impact of the 1957 Glasgow mass tuberculosis screening intervention to inform contemporary approaches to ACF in high tuberculosis burden countries.

## Methods

This study is reported as per the Strengthening the Reporting of Observational Studies in Epidemiology (STROBE) guideline (S1 Checklist).

### Mass miniature X-ray screening intervention

The Glasgow mass X-ray campaign was coordinated by the Corporation of Glasgow, the Western Regional Hospital Board, the Scottish Information Office, and the Department of Health for Scotland, and mass miniature X-ray screening took place between the 11th of March and 12th April 1957 [18]. A total of 37 mobile miniature X-ray units with radiographers were seconded from cities across the UK, and the campaign was supported by approximately 12,000 Glasgow volunteers. Across the city, screening activities were divided into 6 sections that aligned with 5 geographical city divisions plus a further section focused on activities within the city centre and business area.

Prior to the campaign substantial publicity and public engagement was undertaken, including: house-to-house visits within each city ward to distribute X-ray invitation cards; press and cinema advertising; poster displays; church services; distribution of leaflets, banners and stickers; loudspeaker vans and an illuminated tramcart; pavement stencils; aeroplane banner advertising; 2 specially commissioned campaign songs that were played on the radio and at football matches; and information printed on municipal letterheads.

All people who underwent chest X-ray received a badge, and randomly selected people wearing badges within Divisions received small gifts such as chocolates, chickens, and cigarettes. During each week of the campaign, larger gifts (refrigerators, televisions, washing machines, holidays, furniture, and a car) were distributed by selecting X-ray cards at random, and the 100,000th, 200,000th, and 250,000th person X-rayed received additional gifts.

X-ray units were situated in department stores, churches, schools, and other municipal buildings. People aged 15 years or older were invited to receive chest X-ray, regardless of the presence or absence of symptoms, with miniature films interpreted by medical officers. Where there was "significant radiological abnormality requiring further investigation or supervision," people were recalled for a large film at a central X-ray site, and assessment/further investigation by physicians at 5 city hospitals chest clinics (1 per Division, including microbiological testing of sputum) and tuberculosis treatment (if required).

### Setting and population

For this analysis, we used the municipal boundaries of Glasgow City as defined in 1951. Wards are electoral boundaries used in UK national parliamentary elections, and Glasgow City was divided into 37 wards within 5 divisions (North, East, Central, South-East, and South-West). To obtain spatial boundaries, we digitised a scale 1951 Post Office Directory map held in the City of Glasgow Archives using QGIS software (S1 Fig).

We extracted overall city and ward-specific annual population estimates as reported in Medical Officer of Health reports for Glasgow between 1950 and 1963 [19]. In reports, population estimates were based on national censuses (1951 and 1961), updated annually by the Medical Officer of Health based on linear projections of ratios of census populations to registered voters. For this analysis, we used 2 population denominators available: (a) total population (all people identified as usually resident within the City of Glasgow, including people in long-term institutional care and sailors stationed on ships of the Royal Navy at sea or in ports abroad);

and (b) the population excluding people in institutional care or shipping. In reports, population denominators were adjusted for each year to account for city-wide and ward-specific trends. Population estimates stratified by age and sex were not available for each ward.

### Tuberculosis notifications

For each year between 1950 and 1963, we extracted from annual Medical Officer of Health Reports for the City of Glasgow [19] overall numbers of tuberculosis notifications (stratified by pulmonary and extra-pulmonary status, and separately by age group and sex) and the ward-specific notifications (stratified by pulmonary and extra-pulmonary status, and sex). In 1962 and 1963, extra-pulmonary cases were not reported by ward due to small numbers.

### Statistical methods

We calculated the annual pulmonary and extra-pulmonary tuberculosis case notification rate (CNR) for the City of Glasgow using overall numbers of notifications and the total population denominator and scaled these data per 100,000 people. We additionally calculated the ward-specific pulmonary and extrapulmonary CNRs using the population excluding people in institutional care or shipping as the denominator, as cases occurring among institutionalised people or in shipping were not allocated to wards.

To investigate the impact of the 1957 ACF intervention, we constructed multi-level Bayesian interrupted time series regression models (S1 Text) to estimate annual pulmonary tuberculosis CNRs and extra-pulmonary tuberculosis notification rates. Models included terms for a "level-change" in 1957 to capture the immediate effect of the intervention and a "slope change" to estimate changing rates before and after the intervention. To account for over-dispersion, we used the negative binomial distributional family; priors were weakly informative and assessed by inspecting plots of joint prior distributions. Models were fit using the "brms" interface to Stan in R [20], and model convergence was assessed using $\hat{R}$ statistics, effective sample size measures, trace plots of chains, and posterior predictive plots comparing observed data to simulated data from the empirical cumulative distribution function.

We drew 4,000 samples from model posteriors and summarised these (using means and quantile-based 95% uncertainty intervals [UI]). To investigate the impact of the ACF intervention overall and by ward, we predicted counterfactual CNRs for 1958 to 1963 based on a linear projection of trends from the pre-ACF period (1950 to 1956) under the scenario where the ACF intervention had not happened. We then estimated: (i) the relative CNR in 1957 compared to the counterfactual for the same year ("peak effect"); (ii) the "level effect" (relative rate in 1958 versus counterfactual for 1958); and (iii) the "slope effect" (relative annual change in case notifications in 1958 to 1963 compared to the counterfactual scenario). Pairwise correlations between posterior draws for the peak effect, step effect, and slope effect were plotted by ward. Using model predictions, we calculated the number of tuberculosis case notifications that were averted in the post-ACF period (1958 to 1963) compared to that predicted by the counterfactual scenario. We additionally investigated whether the impact of the ACF intervention differed by age and sex by fitting a model to estimate the annual counts of pulmonary tuberculosis case notifications using a Poisson distribution and summarised as above. All analysis was conducted using R version 4.2.3 (R Core Team, Vienna).

### Estimates of case detection and incidence

We estimated the incidence and case detection ratio (CDR; percentage of estimated new pulmonary tuberculosis cases notified annually) for each ward using an equilibrium competing

hazards model and the assumption that the excess notified cases ($N_{during}-N_{pre}$) were prevalent during the intervention and detected with a coverage (*cov*) taken to be 76% based on screening uptake (S2 Text).

$$odds(CDR) = \frac{T \times cov}{(R-1)}$$

Here, $R = N_{during}/N_{pre}$ is the ratio of notification rates during versus before the intervention, $T$ is the average duration of tuberculosis disease in the absence of detection and treatment, taken to be 3 years [21]. We examined the empirical correlations across wards between CDR, pre-ACF incidence, and ACF impact quantified as relative decrease in notifications pre/post-ACF.

### Ethical statement and data availability

Ethical approval was not required for this analysis. Data and code to reproduce analysis are available at https://github.com/petermacp/glasgow-cxr.

## Results

### Population

The total population of Glasgow City was 1.1 million in 1950, declining to just over 1 million by 1963 (Table 1). Ward populations ranged from 16,321 people (Knightswood) to 44,595 (Dalmarnock) in 1950. There was considerable variation in the change in population by 1963, with some wards having substantial population increases (Provan: +312%), and others declining (Exchange: −57%) (S2 Fig). Between 1950 and 1963, we included a total of 14,649,693 person-years of follow-up (population excluding people in institutional care or shipping) in regression models.

**Table 1. Population and tuberculosis notifications in Glasgow City, 1950–1963.**

| Year | Total estimated population* | Pulmonary tuberculosis notifications | Pulmonary tuberculosis case notification rate (per 100,000) | Extra-pulmonary tuberculosis notifications | Extra-pulmonary tuberculosis notification rate (per 100,000) | Total tuberculosis notifications | Total tuberculosis case notification rate (per 100,000) | Percent of tuberculosis notifications that were pulmonary |
|---|---|---|---|---|---|---|---|---|
| 1950 | 1,100,000 | 2,446 | 222.4 | 369 | 33.5 | 2,815 | 255.9 | 86.9% |
| 1951 | 1,089,767 | 2,207 | 202.5 | 355 | 32.6 | 2,562 | 235.1 | 86.1% |
| 1952 | 1,086,800 | 2,264 | 208.3 | 301 | 27.7 | 2,565 | 236.0 | 88.3% |
| 1953 | 1,085,000 | 2,368 | 218.2 | 295 | 27.2 | 2,663 | 245.4 | 88.9% |
| 1954 | 1,084,700 | 2,201 | 202.9 | 241 | 22.2 | 2,442 | 225.1 | 90.1% |
| 1955 | 1,085,100 | 2,181 | 201.0 | 278 | 25.6 | 2,459 | 226.6 | 88.7% |
| 1956 | 1,083,500 | 2,024 | 186.8 | 193 | 17.8 | 2,217 | 204.6 | 91.3% |
| 1957[†] | 1,079,800 | 3,925 | 363.5 | 172 | 15.9 | 4,097 | 379.4 | 95.8% |
| 1958 | 1,078,400 | 1,340 | 124.3 | 167 | 15.5 | 1,507 | 139.7 | 88.9% |
| 1959 | 1,075,800 | 1,159 | 107.7 | 120 | 11.2 | 1,279 | 118.9 | 90.6% |
| 1960 | 1,064,700 | 1,092 | 102.6 | 109 | 10.2 | 1,201 | 112.8 | 90.9% |
| 1961 | 1,053,100 | 1,021 | 97.0 | 137 | 13.0 | 1,158 | 110.0 | 88.2% |
| 1962 | 1,044,500 | 927 | 88.8 | 117 | 11.2 | 1,044 | 100.0 | 88.8% |
| 1963 | 1,029,147 | 863 | 83.9 | 116 | 11.3 | 979 | 95.1 | 88.2% |

*Total populations and case notifications, including those in institutional care and shipping.

[†]Mass miniature X-ray campaign in 1957.

Across each year in Glasgow City, there was a greater number of females than males, with the sex ratio slightly declining from 1:1.10 in 1950 to 1.08 in 1963. The difference in population sex ratio was driven by fewer men compared to women aged 45 years or older (S3 Fig).

### The Glasgow mass miniature chest X-ray campaign

Between 11 March and 12 April 1957, a total of 714,915 people underwent miniature chest X-ray in the campaign. Of these, 19,466 (2.7%) were <15 years, and 73,100 (10.2%) were not resident in Glasgow, leaving 622,349 adult Glasgow-resident participants, 76.0% of the estimated 819,301 adult resident population. A greater percentage of female adult residents (340,474/437,588, 77.8%) than male adult residents (281,875/381,713, 73.8%) underwent chest X-ray, and uptake was highest in younger age groups (S4 Fig). A total of 30,506 people (4.3% of all X-rayed) were recalled for full film, with 652, (2.1%) not attending; 13,900 (45.6%) were subsequently assessed at chest clinics. In total, 2,565 participants were diagnosed as having new active tuberculosis requiring treatment, with 33 of these <15 years and 196 nonresident, resulting in 2,369/622,349 (0.4%) adult Glasgow resident cases detected; approximately 65% were treated as outpatients. A further 1,556 people of all ages were notified with tuberculosis through routine systems during 1957, meaning that the ACF campaign accounted for 60% of all notifications in that year. Of the 2,369 adult Glasgow residents diagnosed with tuberculosis due to screening, 58.5% (1,387) were male, and prevalence was highest in older age groups; 523 (22%) were bacteriologically confirmed by isolation of *M. tuberculosis*. In the year of the campaign, tuberculosis notifications were lower in young children (0 to 5 years) compared to the years before, and a greater percentage of tuberculosis cases were in older age groups; this was maintained in the post-ACF period (S5 Fig). A substantial burden of other disease was identified through the miniature chest X-ray campaign, including lung cancer ($n = 327$), pulmonary fibrosis ($n = 1,279$), and cardiac abnormalities ($n = 1,072$).

### Tuberculosis case notifications prior to active case finding intervention

In 1950, a total of 2,815 people were notified with tuberculosis in Glasgow City, giving a CNR of 255.9 per 100,000. Of these 87% were pulmonary notifications ($n = 2,446$, CNR: 222.4 per 100,000), and 13% were extra-pulmonary ($n = 369$, CNR: 33.5 per 100,000). Pulmonary tuberculosis case notifications rates varied by ward, with the highest CNR in 1950 in Provan (68/19,297, 352.4 per 100,000) and the lowest in Camphill (16/23,630, 67.8 per 100,000)—Fig 1.

Between 1950 and 1956 (pre-ACF period), there was a slow linear decline in pulmonary tuberculosis CNRs overall (equivalent to a 2.3% per year reduction) and in most wards (Fig 1). Wards in the Central and East Districts tended to have higher notification rates than other parts of the city, with some wards showing a flat, or increasing, trend (S6 Fig).

### Impact of active case finding campaign on pulmonary tuberculosis notifications

Inspection of trace plots and model diagnostics showed regression models converged well (S7 Fig and S1 Table).

In 1957, when the mass miniature X-ray active case finding intervention was undertaken, there was a doubling in the CNR for Glasgow City as a whole, increasing from 186.8 per 100,000 in 1956 to 363.5 per 100,000 in 1957. In the interrupted time series regression model, comparing 1957 (ACF intervention year) to the counterfactual scenario there was a 1.95-times (95% UI [1.81, 2.11]) increase in the pulmonary tuberculosis CNR across the city ("peak effect"). At the ward level, there were substantial increases in the pulmonary tuberculosis CNR

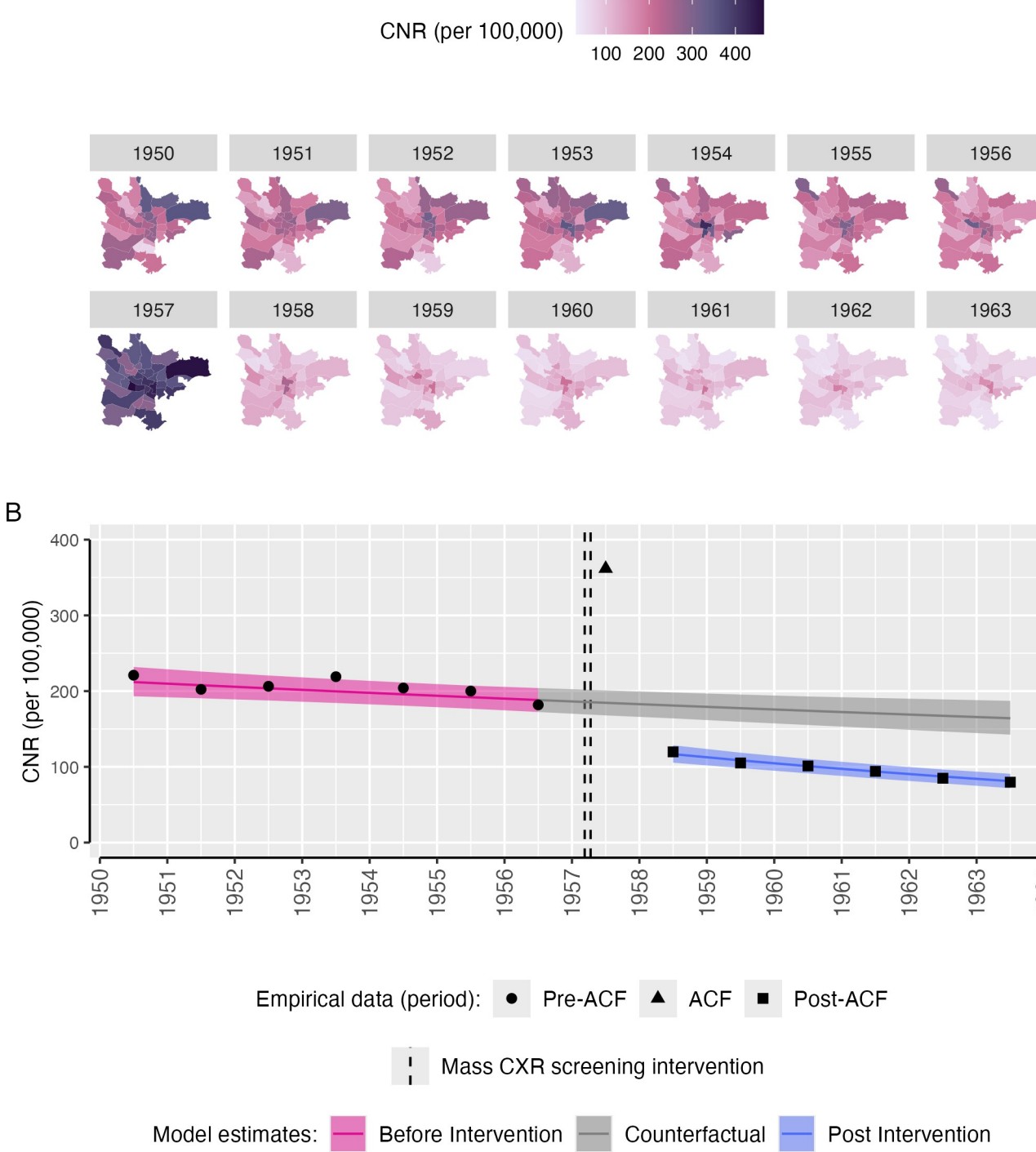

**Fig 1. Pulmonary tuberculosis CNRs in Glasgow City, 1950–1963.** (**A**) Pulmonary tuberculosis CNRs (per 100,000 population) by Glasgow City ward. Ward boundaries obtained from a scale 1951–1952 Post Office Directory map drawn by John Bartholomew FRSG held within the City of Glasgow Archive Special Collections (item PSI-52), digitalised using QGIS 3.34.1. Ward names can be seen in S1 Fig. (**B**) Empirical (points) and modelled (pink and blue bands) pulmonary tuberculosis CNRs (per 100,000 population), with counterfactual of no ACF intervention (grey bands). The mass miniature X-ray active case finding campaign occurred between dashed lines (11 March–12 April 1957). Points are empirical data based on total population estimates and numbers of pulmonary tuberculosis notifications reported to the Glasgow Medical Officer of Health in 1950–1963. Coloured bands are 95% uncertainty intervals, estimated from a Bayesian negative binomial interrupted time-series regression model. ACF, active case finding; CNR, case notification rate.

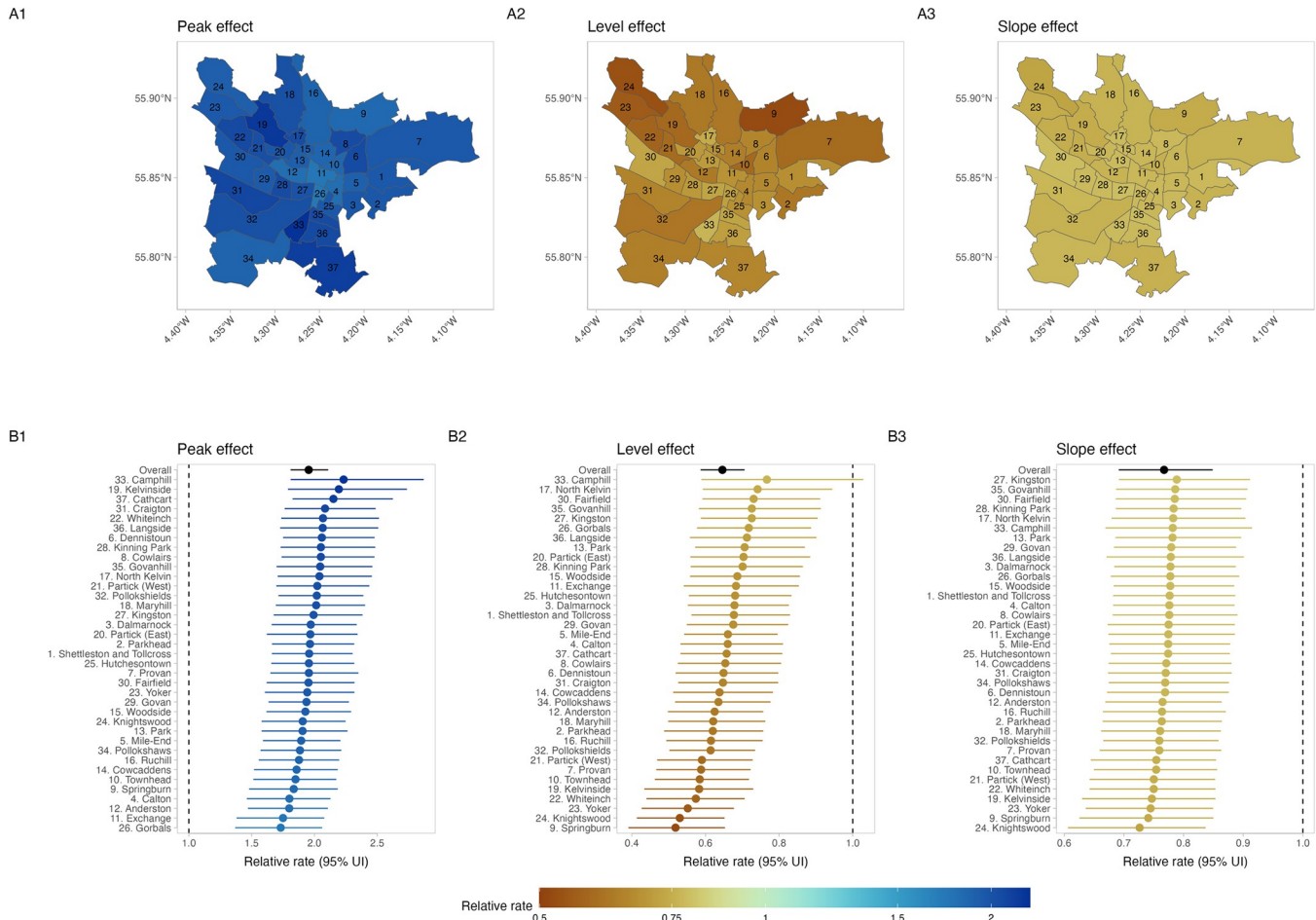

**Fig 2. Impact of mass chest X-ray screening on pulmonary tuberculosis CNRs, overall, and by ward.** (**A1**) Mean posterior relative pulmonary tuberculosis CNR in 1957 vs. counterfactual ("peak effect"). (**A2**) Mean posterior relative pulmonary tuberculosis CNR in 1958 vs. counterfactual ("level effect"). (**A3**) Mean posterior pulmonary tuberculosis relative rate of change in CNRs 1958–1963 vs. counterfactual ("slope effect"). (**A2**) Ward specific and overall (black) "peak effect," with 95% uncertainty interval. (**B2**) Ward specific and overall (black) "level effect," with 95% uncertainty interval. (**A3**) Ward specific and overall (black) "slope effect," with 95% uncertainty interval. UI, uncertainty interval. Ward boundaries obtained from a scale 1951–1952 Post Office Directory map drawn by John Bartholomew FRSG held within the City of Glasgow Archive Special Collections (item PSI-52), digitalised using QGIS 3.34.1.

in all wards, with relative rates ranging from 1.73 (95% UI [1.37, 2.06]) in Gorbals to 2.23 (95% UI [1.81, 2.87]) in Camphill (Fig 2).

There was an overall substantial decline in the pulmonary tuberculosis CNR in the year following the ACF intervention (1958). Comparing 1958 to counterfactual trends, we estimate a 35% relative reduction in the Glasgow City case notification rate (relative rate: 0.65, 95% UI [0.59, 0.71]), with consistent effects across wards (Fig 2). There was a positive correlation between the "peak effect" (increase in CNR in 1957 versus counterfactual) and the "level effect" (decrease in CNR in 1958 versus counterfactual), implying that greater increase in case detection in 1957 was associated with a smaller reduction in case notification in the immediate post-ACF year—S8 Fig.

In the post-ACF period (1958 to 1963), the rate of reduction in pulmonary tuberculosis CNRs was greater than in the pre-ACF period. Overall, across Glasgow City during this 6-year period, there was an annual rate of change of −5.4%, compared to −2.3% in the pre-ACF period. All wards had a markedly greater reduction in pulmonary tuberculosis CNRs ("slope effect") in the post-ACF period compared to the pre-ACF period (Fig 2). There was a positive

**Table 2. Impact of mass screening intervention in post-ACF period (1958–1963) compared to counterfactual scenario.**

| | Year | Difference in case notification rate (per 100,000 py, [95% UI]) | Relative case notification rate [95% UI] | Tuberculosis notifications averted [95% UI] | Percentage difference in cases [95% UI] |
|---|---|---|---|---|---|
| PTB | 1958 | −64.2 [−79.2, −50.1] | 0.65 [0.59, 0.71] | 675.3 [526.5, 832.8] | −35.4% [−41.3%, −29.4%] |
| PTB | 1959 | −68.9 [−84.4, −54.6] | 0.61 [0.56, 0.67] | 723.7 [572.9, 886.1] | −38.8% [−44.3%, −32.7%] |
| PTB | 1960 | −73.1 [−90.1, −57.9] | 0.58 [0.52, 0.64] | 759.1 [601.1, 935.1] | −41.9% [−47.6%, −36.0%] |
| PTB | 1961 | −76.9 [−95.2, −60.2] | 0.55 [0.49, 0.61] | 789.3 [618.6, 977.6] | −44.9% [−50.7%, −38.8%] |
| PTB | 1962 | −80.2 [−99.8, −62.2] | 0.52 [0.46, 0.59] | 817.3 [634.3, 1,017.2] | −47.8% [−53.9%, −41.3%] |
| PTB | 1963 | −83.1 [−104.3, −63.7] | 0.50 [0.43, 0.56] | 834.4 [640.3, 1,047.9] | −50.4% [−56.9%, −43.5%] |
| **PTB** | **Overall (1958–1963)** | | | 4,599.2 [3,641.4, 5,683.4] | −42.9% [−48.7%, −36.9%] |
| EPTB | 1958 | −2.3 [−5.5, 0.8] | 0.86 [0.70, 1.05] | 24.1 [−8.2, 57.4] | −13.7% [−30.2%, 5.4%] |
| EPTB | 1959 | −1.8 [−4.8, 0.9] | 0.88 [0.72, 1.07] | 18.9 [−9.1, 50.1] | −11.6% [−28.2%, 6.8%] |
| EPTB | 1960 | −1.4 [−4.5, 1.3] | 0.91 [0.72, 1.12] | 14.0 [−13.9, 46.4] | −9.3% [−28.1%, 11.8%] |
| EPTB | 1961 | −1.0 [−4.3, 2.1] | 0.93 [0.70, 1.20] | 9.8 [−21.4, 44.2] | −6.7% [−29.7%, 20.0%] |
| EPTB | **Overall (1958–1963)** | | | 66.8 [−42.7, 192.2] | −10.6% [−27.9%, 8.7%] |

All estimates are compared to counterfactual scenario where active case finding intervention had not been implemented.

PTB, pulmonary tuberculosis; EPTB, extra-pulmonary tuberculosis; py, person-years.

correlation between greater "peak effect" of ACF, and lower relative reductions in the rate of change post-ACF ("slope effect"). There was no correlation between the "level effect" and the "slope effect"—S8 Fig.

In our extra-pulmonary tuberculosis model, the ACF intervention had little discernible peak (RR: 0.89, 95% UI [0.72, 1.09]), level (RR: 0.85, 95% UI [0.69, 1.03]), or slope effect (RR: 1.13, 95% UI [0.77, 1.60]), either across the city, or at ward level (S9 and S10 Figs).

## Estimates of case detection

From our competing hazards model, in the pre-ACF period (1950 to 1956), we estimate that prevalence was <400 per 100,000 in all wards. Assuming a 76% ACF coverage (as estimated from the total number screened and the estimated adult resident population), we estimated ward case detection ratios ranging from 60% to 92%, with a median (IQR) of 78% (72%, 82%). Across wards, we found pre-ACF tuberculosis CNR and case detection ratios were positively correlated (corr = 0.76), and case detection ratios were negatively correlated with ACF impact measured as pre/post notification ratio (corr = −0.55)—S11 Fig.

## Cases averted compared to counterfactual scenario

Across the City of Glasgow between 1958 and 1963, we estimate that, compared to the counterfactual scenario where the ACF intervention had not happened, there were 4,599 (95% UI [3,641, 5,683]) pulmonary tuberculosis notifications averted, equivalent to a 42.9% (95% UI [36.9%, 48.7%]) reduction (Table 2).

## Impact of active case finding by age group and sex

In the pre-intervention period, there were differences in pulmonary tuberculosis case notification trajectories by age group and sex, with in general declining rates in younger age groups,

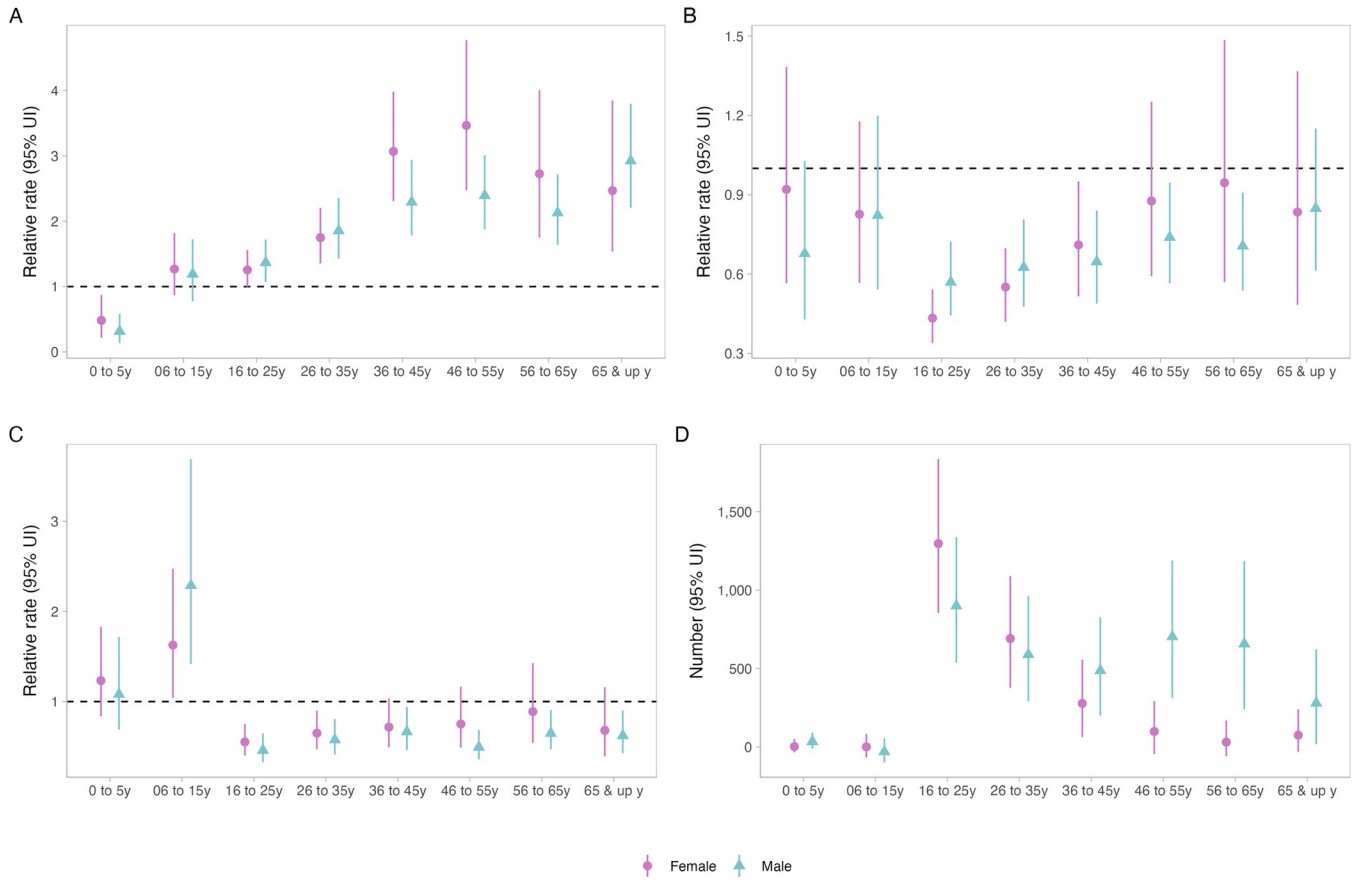

**Fig 3. Impact of active case finding on case notifications by age group and sex, Glasgow City: 1950–1963.** (**A**) "Peak effect" on case notifications (1957 vs. counterfactual). (**B**) "Level effect" on case notifications (1958 vs. counterfactual). (**C**) "Slope effect" on case notifications (1958–1963 vs. counterfactual). (**D**) Numbers of pulmonary tuberculosis notifications averted during 1958–1963, compared to counterfactual scenario. UI, uncertainty interval.

higher numbers of notifications among older men than women, and older men in particular having an upwards trend in notifications, compared to other groups (S12 Fig). With the intervention, there was a reduction in case notifications in young children (0 to 5 years) (Fig 3). The greatest difference between men and women in cases averted was in older age groups.

## Discussion

The Glasgow mass miniature X-ray campaign for tuberculosis in 1957 was probably the largest single community-based active case finding intervention ever undertaken, with more than 715,000 people (622,349 adult Glasgow residents; 76% of the entire adult resident population) screened with miniature chest X-ray over just 5 weeks, equating to nearly 150,000 people screened per week of the campaign. In our analysis using modern epidemiological methods, we found that the mass screening intervention had a substantial immediate impact—doubling case notifications—and resulted in more rapid reductions in the citywide notification rate (5.4% per year) post intervention compared to the pre-intervention period (2.3%), with an estimated 4,599 notified pulmonary tuberculosis case notifications averted between 1958 and 1963 compared to the counterfactual scenario. Consistency in intervention effect across city wards gives confidence that the impact can be attributed to the intervention. By mostly omitting historical evidence of the impact of well-conducted mass X-ray screening programmes in

global guideline development and contemporary discourse, it is likely that the potential impact of intensive, high-coverage ACF interventions to change epidemic trajectories has been underestimated.

Neither systematic reviews of historical ACF interventions [14,22,23] nor WHO guidelines for systematic tuberculosis screening [12,13] have included these data from Glasgow in evidence synthesis. A review of historical mass miniature X-ray screening interventions from the 1930s to late 1960s identified 18 published reports from North America, Europe, and India [6]. Synthesising these data Golub and colleagues concluded that mass miniature X-ray screening was highly effective at increasing case detection, particularly when programmes were well supported by community and health system engagement [6]. Likewise, Miller and colleagues emphasised the importance of logistics at large scale to ACF programme success [24]. During the late 1950s however, several large cities in the UK—including Glasgow, Edinburgh, and Liverpool, as well as many cities and countries internationally (Brazil, Japan, Spain, Italy, Sweden) undertook mass miniature X-ray screening for tuberculosis. However, to the best of our knowledge, evidence from these programmes have mostly not been collated, possibly because data reside only in municipal public health reports, rather than in more accessible scientific publications. Moreover, data included in both sets of WHO guidance for community-based systematic screening (2013 [12] and 2021 [13]) include only a small number of historical studies. This represents a missed opportunity: many high tuberculosis burden countries have tuberculosis epidemiological characteristics similar to Europe and North America in the 1950s. By omitting historical evidence, there is a danger that previous lessons learned are forgotten.

Only a small number of contemporary studies have investigated the impact of ACF for tuberculosis on CNRs, with the large majority following up communities only in the period before and during the intervention [14], and with nearly all showing large increases in case detection during implementation. In non-randomised studies, following up communities for prolonged periods after the ACF intervention is critical to identify effects that are temporary or short lived; our analysis here provides compelling evidence that a single intervention with high coverage can change epidemic trajectory. Two contemporary randomised trials of community-based active case finding (ACT3 [15] and TREATS [16]) have shown mixed results on tuberculosis prevalence and infection, with the more intensive ACT3 study in Vietnam resulting in a substantial reduction in prevalent pulmonary tuberculosis following universal sputum testing with Xpert, whereas no effect was identified from the symptom screening approach in TREATS. We speculate that the very high coverage of chest X-ray screening achieved in Glasgow in 1957 was a major contributor to epidemiological impact, and potentially identified people with early and subclinical tuberculosis, rapidly reducing transmission. This would align with the experiences of the ACT3 trial, where universal sputum testing, as in the Glasgow mass screening campaign, likely identified people in the subclinical state [25]. We additionally found that, in the pre-ACF period (1950 to 1956) all Glasgow wards had an estimated tuberculosis prevalence of <400 per 100,000. This is below the current WHO screening threshold of >500 per 100,000 and suggests that epidemiological impact from ACF may be achieved in settings with more concentrated epidemics; this needs confirmation through analysis of a greater number of historical and contemporary data sets. Further research to understand the epidemiological impact of the effect of detection and treatment of early states of tuberculosis is also needed.

Critically important to achieving high population coverage of tuberculosis screening was a programme of mass community engagement and mobilisation, supported by more than 12,000 volunteers. A recent qualitative synthesis of the community views of participating in ACF programmes emphasises that local ownership and leadership, ongoing support for people

screened, and health systems strengthening to support the increased healthcare demands generated by mass screening are important determinants of intervention success [26]. The Glasgow mass screening campaign exemplified these principles, and we argue that community and health systems support within contemporary ACF programmes are often insufficient, and delivered in a "top-down"—rather than community-responsive—fashion, likely contributing to lower than anticipated participation and effectiveness.

The postwar period between 1948 and 1960 was a period of tremendous social change, and the 1957 Glasgow mass tuberculosis screening campaign needs to be contextualised alongside progressive improvements in living conditions, nutrition (with the ending of second world war food rationing in 1954), healthcare, and tuberculosis care and prevention. In Glasgow, "slum clearances" commenced in the mid-1950s, particularly in wards in the centre and south of the city such as Camphill, Gorbals, Cowcaddens, and Govan, with new residential schemes established on the peripheries of the city. These changes in ward demographics can be seen in our supplementary figures. However, while the reduction in crowding and removal of dwellings in the worst condition is likely to have beneficial effects for reducing tuberculosis transmission, the slow improvements in social and housing conditions themselves are unlikely to account for the rapid and large changes in tuberculosis notifications that occurred immediately with and in the period after the mass screening campaign.

According to historical Medical Officer of Health reports, Bacillus Calmette–Guérin (BCG) vaccination was first introduced in Glasgow in 1950 and offered to people with a positive tuberculin skin test and in one of the following four categories: "nurses in hospitals, especially institutions for tuberculosis"; "newborn infants of tuberculous mothers"; "contacts of cases of open tuberculosis"; and medical students. In September to November 1953, this BCG vaccination programme was extended to include school children aged 13 years with a positive tuberculin skin test. During the mass screening campaign in 1957, tuberculin skin test positive contacts of participants diagnosed with tuberculosis were additionally offered BCG vaccination. Although systematic reviews suggest that there is evidence to support a protective effect of BCG vaccination on infection and disease, particularly for younger children [27,28], the greatest protective benefit appears to be to infants at risk of severe tuberculosis disease, who themselves are unlikely to transmit to others. Overall, we believe that the Glasgow BCG vaccination programme was unlikely to have been a major contributor to tuberculosis control efforts and is unlikely to explain the substantial increase in the rate of decline of tuberculosis CNRs immediately following the mass screening programme. However, we acknowledge that the combined effects of screening, social improvements, and improved tuberculosis diagnosis and treatment remain difficult to untangle.

We found evidence that the impact of the ACF intervention differed by age group and sex. In young children (<5 years, who themselves were not eligible for screening, although a small number of children <15 years did undergo chest X-ray), CNRs decreased during the intervention year in contrast to other age groups which saw large increases; we speculate that this may indicate evidence of ACF shortening infectious duration and providing early beneficial impact on transmission. For adults, contemporary data from high tuberculosis burden countries show that men have twice the prevalence of undiagnosed pulmonary tuberculosis than women [29]. Men are less likely to participate in prevalence surveys in Africa [30] and Asia [31], and in active case finding trials [15,16,32], prompting calls to target interventions towards men [17]. However, if tuberculosis prevalence was indeed higher in men than women in Glasgow in the 1950s, our analysis suggests that rapid delivery of high coverage of high sensitivity screening (with chest X-ray)—rather than interventions targeted specifically at men—may achieve benefits for all groups, but especially those where disease burden is likely to have been highest. We found no effect on extrapulmonary tuberculosis; this is to be expected, as extrapulmonary

tuberculosis notification rates were already low by the time the intervention started, and mass chest X-ray screening targets pulmonary tuberculosis. Importantly, our analysis could only investigate population-level effects on case notifications; previous reviews have emphasised the dearth of information on the individual-level benefits and harms of participating in community tuberculosis screening [33]. There may additionally be important implications for health systems when planning mass screening campaigns. In the Glasgow campaign, additional chest clinics were required to run during weekdays and at weekends to meet demands of new referrals from the screening programme. Although mass tuberculosis screening campaigns may be an opportunity to integrate combined health and public health surveillance interventions (as we have previously argued [17]), programmes need to plan for the substantial additional healthcare resources that will likely be required due to detection of other health issues; in the Glasgow campaign, a substantial burden of non-tuberculosis pulmonary disease was identified requiring assessment at city hospital chest clinics. More recent data from Kenya [34] and South Africa [35] emphasises that in countries undergoing health and demographic transitions, the health needs and prevalence of noncommunicable diseases in people participating in tuberculosis screening programmes continues to be high.

There was little description available of microbiological testing results or treatment outcomes, and it is possible that there was overtreatment of tuberculosis in people screened, or indeed treatment of people with very early subclinical tuberculosis which would not have usually been detected and treated. It is unclear what the implications of potential overtreatment, or indeed treatment of early tuberculosis, would be for participants and health systems. Future ACF trials should systematically record individual- and health systems-benefits and harms of participating in tuberculosis screening programmes, and greater research is needed into the effects of screening and treatment of early tuberculosis disease.

Despite the high-quality data available and the rigorous epidemiological methods used, there are some limitations to this analysis. We assumed a linear trend in tuberculosis CNRs in the pre-ACF period, and projected this forward for the counterfactual scenario in the post-ACF period. It is possible however that—in the absence of the intervention—the rate of decline may have accelerated downwards in the late 1950s to early 1960s, leading us to overestimate impact. We relied on available census data and official ward population estimates from government and municipal sources; however, the population of Glasgow changed rapidly in the late 1950s, and these may have over- or underestimated population denominators, particularly for some wards. We did not have comprehensive data on microbiological results or treatment outcomes for notified cases throughout the study period, nor for participants in the intervention; a post-intervention study of prevalence was not done. Finally, while the rapidity, magnitude, and consistency of impact across wards strongly support a direct causal effect, other factors ("slum clearances," mobility, general improvements in living conditions, access to healthcare, nutrition, and air quality) may have contributed to improvements in tuberculosis epidemiology.

In conclusion, rapid mass miniature chest X-ray screening of over 715,000 people in Glasgow, Scotland (76% of the entire adult city population within 5 weeks) resulted in an accelerated and sustained reduction in tuberculosis CNRs in Glasgow, with an estimated 4,599 pulmonary tuberculosis case notifications averted in the 6 years following the intervention, and potentially evidence of an intervention effect on reducing transmission to young children. Previous attempts to synthesise data from tuberculosis mass screening programmes have mostly ignored examples from the 1950 to 1960s held in municipal health departments, with the consequence that our understanding of the population benefit of ACF has likely been underestimated. Synthesis and comparative analysis of other historical mass tuberculosis screening programmes could give insights into the magnitude of effectiveness of community-

based active case finding programmes. Cities in today's high tuberculosis burden countries may benefit from mass chest X-ray screening supported by high levels of health system and community engagement, alongside improvements in living conditions.

## Supporting information

**S1 STROBE Checklist. STROBE statement.**
(DOCX)

**S1 Table. Summary of posterior draws from pulmonary tuberculosis model.**
(CSV)

**S1 Text. Additional methods.**
(DOCX)

**S1 Fig. Divisions and Wards of the City of Glasgow, 1951.** Red box in inset map of Scotland shows location of the main figure (City of Glasgow). Map of Scotland from the UK Office for National Statistics Open Geography Portal (https://geoportal.statistics.gov.uk/datasets/ons:: countries-december-2023-boundaries-uk-bfc-2/about), licensed under the Open Government Licence v.3.0 (https://www.nationalarchives.gov.uk/doc/open-government-licence/version/3/ ). City of Glasgow ward boundaries obtained from a scale 1951–1952 Post Office Directory map drawn by John Bartholomew FRSG held within the City of Glasgow Archive Special Collections (item PSI-52), digitalised using QGIS 3.34.1.
(TIFF)

**S2 Fig. Glasgow City population by ward.**
(TIFF)

**S3 Fig. Glasgow City population pyramids by year.**
(TIFF)

**S4 Fig. Uptake of tuberculosis screening by age and sex.**
(TIFF)

**S5 Fig. Distribution of pulmonary tuberculosis cases by age group and sex in Glasgow by study period.** ACF: active case finding.
(TIFF)

**S6 Fig. Pulmonary tuberculosis case notification rates by Glasgow Ward, 1950–1963.** Empirical and modelled case notification rates (per 100,000 population) by ward, with counterfactual of no active case finding intervention. The mass miniature X-ray active case finding campaign occurred between dashed lines (11th March–12th April 1957). CNR: case notification rate. ACF: active case finding.
(TIFF)

**S7 Fig. Rank plots of Markov chain Monte Carlo draws from pulmonary tuberculosis model.**
(TIFF)

**S8 Fig. Posterior distributions of, and correlations between "peak effect," "level effect," and "slope effect" of active case finding intervention.**
(TIFF)

**S9 Fig. Impact of active case finding intervention on extra-pulmonary tuberculosis case notification rates, Glasgow City.** Empirical and modelled case notification rates (per 100,000

population) by ward, with counterfactual of no active case finding intervention. The mass miniature X-ray active case finding campaign occurred between dashed lines (11th March–12th April 1957). CNR: case notification rate. ACF: active case finding.
(TIFF)

**S10 Fig. Impact of active case finding intervention on extra-pulmonary tuberculosis case notification rates, ward-level.** Empirical and modelled case notification rates (per 100,000 population) by ward, with counterfactual of no active case finding intervention. The mass miniature X-ray active case finding campaign occurred between dashed lines (11th March–12th April 1957). CNR: case notification rate. ACF: active case finding.
(TIFF)

**S11 Fig. Correlation between estimated case detection active case finding impact.** Points are Ward specific values. CDR: Case detection rate; RR.level: Mean posterior relative pulmonary tuberculosis case notification rate in 1958 vs. counterfactual ("level effect"); Ipre: mean estimated incidence per 100,000 in pre-ACF period; Npre: mean case notification rate per 100,000 in pre-ACF period; prev: estimated prevalence per 100,000 in pre-ACF period (1950–1956); pdens: population density (1,000 people per square kilometre).
(TIFF)

**S12 Fig. Impact of active case finding intervention on pulmonary tuberculosis by age and sex.** Empirical and modelled case notification rates (per 100,000 population) by ward, with counterfactual of no active case finding intervention. The mass miniature X-ray active case finding campaign occurred between dashed lines (11th March–12th April 1957). CNR: case notification rate. ACF: active case finding.
(TIFF)

## Acknowledgments

We gratefully acknowledge the staff of the Special Collections at the City of Glasgow Archives for facilitating access to historical reports and maps. We also acknowledge Callan T MacPherson and Isabella L MacPherson for assistance in data extraction from historical records.

## Author Contributions

**Conceptualization:** Peter MacPherson, Helen R. Stagg, Alvaro Schwalb, Hazel Henderson, Alice E. Taylor, Rachael M. Burke, Hannah M. Rickman, Cecily Miller, Rein M. G. J. Houben, Peter J. Dodd, Elizabeth L. Corbett.

**Data curation:** Peter MacPherson, Peter J. Dodd.

**Formal analysis:** Peter MacPherson, Helen R. Stagg, Peter J. Dodd.

**Investigation:** Helen R. Stagg, Alice E. Taylor, Rachael M. Burke, Rein M. G. J. Houben, Peter J. Dodd, Elizabeth L. Corbett.

**Methodology:** Peter MacPherson, Helen R. Stagg, Rachael M. Burke, Hannah M. Rickman, Cecily Miller, Rein M. G. J. Houben, Peter J. Dodd, Elizabeth L. Corbett.

**Project administration:** Elizabeth L. Corbett.

**Supervision:** Elizabeth L. Corbett.

**Visualization:** Peter J. Dodd.

**Writing – original draft:** Peter MacPherson, Hannah M. Rickman, Peter J. Dodd, Elizabeth L. Corbett.

**Writing – review & editing:** Peter MacPherson, Helen R. Stagg, Alvaro Schwalb, Hazel Henderson, Alice E. Taylor, Rachael M. Burke, Hannah M. Rickman, Cecily Miller, Rein M. G. J. Houben, Peter J. Dodd, Elizabeth L. Corbett.

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
