## [Editor Report · Decision Letter 0]

22 Jul 2024

Dear Dr MacPherson, 

Thank you for submitting your manuscript entitled "Impact of active case finding for tuberculosis with mass chest X-ray screening in Glasgow, Scotland, 1950-1963: an epidemiological analysis of historical data" for consideration by PLOS Medicine.

Your manuscript has now been evaluated by the PLOS Medicine editorial staff as well as by an academic editor with relevant expertise and I am writing to let you know that we would like to send your submission out for external peer review.

Please re-submit your manuscript within two working days, i.e. by Jul 24 2024 11:59PM.

Feel free to email me at kjanin@plos.org if you have any queries relating to your submission.

Kind regards,

Katrien G. Janin, PhD

Senior Editor

PLOS Medicine

---

## [Decision Letter · Decision Letter 1]

29 Aug 2024

Dear Dr MacPherson,

Many thanks for submitting your manuscript "Impact of active case finding for tuberculosis with mass chest X-ray screening in Glasgow, Scotland, 1950-1963: an epidemiological analysis of historical data" (PMEDICINE-D-24-02288R1) to PLOS Medicine. The paper has been reviewed by subject experts and a statistician; their comments are included below and can also be accessed here: [LINK]

As you will see, the reviewers [ADD DETAILS AS APPROPRIATE]. After discussing the paper with the editorial team and an academic editor with relevant expertise, I'm pleased to invite you to revise the paper in response to the reviewers' comments. We plan to send the revised paper to some or all of the original reviewers, and we cannot provide any guarantees at this stage regarding publication.

We ask that you submit your revision by Sep 19 2024 11:59PM. However, if this deadline is not feasible, please contact me by email, and we can discuss a suitable alternative.

Don't hesitate to contact me directly with any questions (kjanin@plos.org). 

Best regards, 

Katrien 

Katrien Janin, PhD 

Associate Editor

PLOS Medicine

kjanin@plos.org

Comments from the academic editor:

Thanks for this interesting article. We would need the authors to temper the language a bit though, although epidemiologically the inidence rates are similar in some LMIC now to where Scotland was in the 50s, the diagnostics used in ACF and (treatment available) is very different. So the generalisability will be fairly limited. 

Editorial comment:

On the practices of the times, we wondered how the BCG vaccination may have played a role. If online sources are to be believed, the vaccination program started in 1953 and we wondered if (and how) this may have an impact. 

Comments from the reviewers: 

Reviewer #1: Thank you for the excellent work.

The topic, methodology and outcome from your study will have relevance to many high burden settings. 

This work presents the opportunity to inform policy while built on the usage of existing data.

Active case finding for TB and the use of CXRs are a very important topic as we try and address the substantial burden of undiagnosed TB.

Comments:

Line 123-126: Great to include details on the incentives used in the implementation. Line 405: you do speculate around the role of the high coverage of the screening

It may be worthwhile revisiting this in the discussion with some consideration of what it may take to achieve similar in our current context. The use of incentives in trials has substantial oversight from ethics committees and it is near impossible for me to conceive that an existing Ministry of Health in a high burden setting would be able to budget and implement such an incentive programme. 

In addition, the irony of cigarette incentives for a chest x ray screening programme warrants some consideration of the evolution in Public Health approach that the world has seen in the last 70 years.

Line 253-254: This nicely points to potential value of CXR screening beyond TB. It is understandable that more detailed analysis of this is outside of the scope of your article, but a comment in the discussion on this would be very useful. As high burden countries in resource limited settings need to consider costs and benefits, the utliity of this programme beyond TB could further support implementation. Perhaps reference to any other work that has projected the additional benefits and opportunities to integrate with more holisitic health screening or the need for future analyses that explore this would be good

During the ACF intervention - Glasgow almost doubled its case notification in that 1 year. While you have stated you were unable to track and report outcomes, it is essential to describe and consider the ability of a City to effectively deal with a doubling in case notification. In the conclusion you describe the support needed for the implementation of the CXR screening but weak/fragile health systems could be especially vulnerable to a surge in TB diagnoses. Could you consider describing what Glasgow had in place to manage this increase or what a high-burden, resource limited setting may want to put in place to ensure optimum treatment outcomes

Minor comments

Line 55-56: Consider changing to month by name as per main text to eliminate confusion with ddmmyyyy and mmddyyyy

Line 389: American should be America

Line 389: missing is .....evidence, there is a danger.........

line 471: delete of

Thank you

Reviewer #2: This represents a key addition to the historical evidence base that community-wide systematic screening.

Crucially, current WHO TB screening recommendation offer only a conditional recommendation, when prevalence is 0.5% or greater. I totally concur with the premise of the paper, that the historical evidence in general has been insufficiently evaluated to inform current screening recommendations and practices. 

This paper documents a historical experience where prevalence was demonstrably less than that threshold, and documents the efforts and impact of a single campaign intervention fairly exhaustively.

It represents a remarkable contribution to the historical evidence base, complements the re-analysis from Kolin that many of the authors have participated in previously, and demonstrates that an analysis 50 years late is still better late than never. 

My first reaction was to just accept the paper. But a few nagging issues have made me change to minor revision. There are some unanswered questions, which I would encourage the authors to explore. 

major issue 1: The issue of poor bacteriological confirmation during the screening and the lack of a sub-analysis around microbiologically confirmed cases. 

This depends on the quality of bacteriology available during the day, and it's utilization, which is not quite addressed. 

The paper mentions practices, "where there was "significant radiological abnormality requiring further investigation or supervision",

132 people were recalled for a large film at a central X-ray site, and assessment/further investigation

133 by physicians at five city hospitals chest clinics (one per Division, including microbiological

134 testing of sputum) and tuberculosis treatment (if required).

What does that microbiological examination consist of? Is there data?

We are told that "A total of 30,506 people

241 (4.3% of all X-rayed) were recalled for full film, with 652, (2.1%) not attending; 13,900 (45.6%)

242 were subsequently assessed at chest clinics.

Later we are told that in the screening program

 2,369/622,349 (0.4%) Glasgow resident cases detected; approximately

245 65% were treated as outpatients. A further 1,556 people of all ages were notified with

246 tuberculosis through routine systems during 1957, meaning that the ACF campaign accounted

247 for 60% of all notifications in that year. Of the 2,369 adult Glasgow residents diagnosed with

248 tuberculosis due to screening, 58.5% (1,387) were male, and prevalence was highest in older

249 age groups; 523 (22%) were bacteriologically-confirmed by isolation of M. tuberculosis. 

\\With only 22% of detected cases that were bacteriologically confirmed, one wonders, what exactly was being detected and treated? Amongst these 13,900 detected by the screening program and referred to chest clinics with full radiographs, 4 in 5 were unable to be bacteriologically confirmed, by the 

How important, or not important these individuals were to the subsequent impact observed remains untold, and indeed unexplored. Because there were still 1,556 persons routinely diagnosed during the year, presumably because they were sick. One could argue that it doesn't matter, because the screening program happened, they detected and treated these cases, and that was the impact. But because the intervention was mass radiographic screening, involving 37 units and 12,000 volunteers and a whole of society campaign for a paltry 1 million population (one unit and ~300 volunteers for every ~25K population), it does matter. Was the additional impact that is shown very convincingly being driven by the 523 who were (presumably) smear or culture positive, vs the (2369-523=)1846 who were apparently bacteriologically negative? Was the reduction in level and slope of notification a direct effect of removing prevalent TB cases and their subsequent transmission, or was it an effect of 'treatment as prevention' to radiographic abnormals, effectively pre-treated a subset of future cases in the campaign year? Was the effect due to reduced transmission, or due to mass prophylactic treatment to a thousands of high risk individuals?

Accordingly I'd recommend (a) the notification table 1 (routine and screening) be updated and split by bacteriologic confirmation (or an additional main table added with the stratification - not the supplement, please). (b) a sub-analysis accounting for the pre and post screening impact on bacteriologically confirmed TB be conducted. I don't know if it will have the power to detect. This recommendation is contingent on the bacteriological practices of the day. If the bacteriological confirmation was done by smear only, then this is probably not worth it to do the subanalysis due to insensitivity of detection, and would just add the caveat around bacteriologic confirmation as a limitation in the discussion. 

There's some reference in the discussion that this might not have been possible. 

"There was little description available of microbiological

454 testing results or treatment outcomes, and it is possible that there was over-treatment of

455 tuberculosis in people screened, or indeed treatment of people with very early subclinical TB

456 which would not have usually been detected and treated. "

However, I'd suggest that this be included in the methods up front and at least better discussion of what it was that actually might have been driving change. 

Minor issue 1: 

"However, uptake of screening was low in these two trials, with only 45% of eligible

404 participants screened by sputum Xpert in the best performing year of ACT3. We speculate that

405 the very high coverage of chest X-ray screening achieved in Glasgow in 1957 was a major

406 contributor to epidemiological impact, and potentially identified people with early and

407 subclinical tuberculosis, rapidly reducing transmission."

I have some issue with this interpretation. In ACT3, nearly 80% of the ennumerated population was reached, but specimens were only successfully collected and tested from 45%. To declare that this was low coverage and that's the possible difference is to infer that the prevalence of disease was similar amongst those reached who could not produce a sputum vs those who were able to produce a sputum. That is not consistent with data from prevalence surveys. Also ACT3 had very minimal criteria to reject a sputum, basically volume based. Also ACT3 had a relatively large impact by the metrics provided. So the representation of ACT3, and the point being made about it, doesn't make sense. I think you're trying to say that the effect of ACT3 was actually similar of that observed here, and those differences are interesting and should be explored. On one hand you have a lower sensitivity screening tool (MMR, which is lower than current dCXR can detect), on the other you have molecular testing irrespective of CXR results, including detection of some individuals who may have not had detectable CXR abnormalities. 

So it's apples and oranges, and I think maybe a bit more nuance about the comparison is warranted. 

Minor issue 2:

Discussion - context needed about secular trends, with more information and a bit more humility in attribution of the changes occurring to the screening campaign. 

There's no mention of a rather momentous event of the time, which is the post-war lifting of food rations in 1954 in England, or the effect of improved nutritional conditions. 

That may have had some interaction with the accelerated decline in rates. Which speaks to the question, this was a singled-ended one time event Glasgow. What about places which didn't have such a massive screening campaign, and had to exist with the secular changes of the time? A more thorough evaluation would at least acknowledge this limitation. 

It's mentioned in the limitation:

"It is possible however that - in the absence of the intervention -

463 the rate of decline may have accelerated downwards in the late 1950s to early 1960s, leading us

464 to over-estimate impact."

However, why not actually go farther and propose a more detailed evaluation of the historical evidence of the day, then, comparing other areas? There's a grad student in there, looking for a PhD topic...

Minor issue 3:

"435 We found strong evidence that the impact of the ACF intervention diYered by age group and sex.

436 In young children (<5 years, who themselves weren't eligible for screening, although a small

437 number of children <15 years did undergo chest X-ray), case notification rates decreased during

438 the intervention year in contrast to other age groups which saw large increases; we speculate

439 that this may indicate evidence of ACF shortening infectious duration and providing early

440 beneficial impact on transmission."

This is a bit hard to swallow from the data presented, which appear to have been drawn from very small numbers, for the young children argument. the numbers averted suggested that the rate change was applied to a very small number of cases both before and after the intervention. Perhaps please provide the numbers of notification, and the rate among kids, in the pre and post period. Birth rates may have also been expanding in the postwar era, inflating the infant denominator. 

Perhaps this also needs to be couched with the unmentioned counterpoint, that there was an increase in slope of notifications in the 6-15 age group. So it's not really clear what's 'strong' evidence here. Possible evidence, maybe. 

The conclusion paragraph seems to acknowledge this and pull back. "479 and potentially evidence of an intervention eYect on reducing transmission to young children."

Minor issue 4: 

"475 In conclusion, rapid mass miniature chest X-ray screening of over 715,000 people in Glasgow,

476 Scotland (76% of the entire adult city population within 5 weeks) resulted in an accelerated and

477 sustained reduction in tuberculosis case notification rates in Glasgow, with an estimated 4,656

478 pulmonary tuberculosis case notifications averted in the six years following the intervention,

479 and potentially evidence of an intervention eYect on reducing transmission to young children."

This conclusion may need to be couched in less certain causality, given the uncertainties in secular trends. It's likely to have resulted. 

Lastly I want to thank the authors for taking this analysis on, and for the scholarship involved to recover this learning opportunity from the dustbin of history. It's relevant and important, and PlosMED should publish it ASAP. 

Puneet Dewan, BMGF

Reference: https://assets.publishing.service.gov.uk/media/5a81669340f0b62305b8ebfc/Domestic_Food_Consumption_and_Expenditure_1956.pdf

Reviewer #3: See attachment

Michael Dewey

---

* Please upload any figures associated with your paper as individual TIF or EPS files with 300dpi resolution at resubmission; please read our figure guidelines for more information on our requirements: http://journals.plos.org/plosmedicine/s/figures. While revising your submission, please upload your figure files to the PACE digital diagnostic tool, https://pacev2.apexcovantage.com/. PACE helps ensure that figures meet PLOS requirements. To use PACE, you must first register as a user. Then, login and navigate to the UPLOAD tab, where you will find detailed instructions on how to use the tool. If you encounter any issues or have any questions when using PACE, please email us at PLOSMedicine@plos.org.

* Abstract: Please structure your abstract using the PLOS Medicine headings (Background, Methods and Findings, Conclusions). Please combine the Methods and Findings sections into one section. In the last sentence of the Abstract Methods and Findings section, please describe the main limitation(s) of the study's methodology.

FIGURES AND TABLES

SUPPLEMENTARY MATERIAL

REFERENCES

---

## [Decision Letter · Decision Letter 2]

29 Sep 2024

Dear Dr. MacPherson,

Many thanks for submitting your revised manuscript "Impact of active case finding for tuberculosis with mass chest X-ray screening in Glasgow, Scotland, 1950-1963: an epidemiological analysis of historical data" (PMEDICINE-D-24-02288R2) to PLOS Medicine.

I have discussed the paper with my colleagues and the academic editor, and it was also seen again by the statistical reviewer. I am pleased to say that we were all pleased with the edits to the manuscript and the additional discussion and historical context that was added. As such, we plan to accept the paper for publication in the journal, pending attention to a few remaining editorial and production issues.

The remaining issues that need to be addressed are listed at the end of this email. Any accompanying reviewer attachments can be seen here: [LINK]

As a reminder, we ask every co-author listed on the manuscript to fill in a contributing author statement. If any of the co-authors have not filled in the statement, we will remind them to do so when the paper is revised. If all statements are not completed in a timely fashion this could hold up the re-review process. Should there be a problem getting one of your co-authors to fill in a statement we will be in contact. 

We expect to receive your revised manuscript within 1 week (Oct 7th). If this deadline does not seem feasible, or you have any questions, please email me directly at hvanepps@plos.org. Otherwise, I look forward to receiving the final revision.

Kind regards,

Heather

Heather Van Epps, PhD

Executive Editor 

PLOS Medicine

hvanepps@plos.org

Requests from Editors:

1. Author summary: Please consider revising the third bullet point in the ‘What did the authors do and find’ section as follows (adding ‘rates’ before ‘doubled’ and adding a comma after ‘Post-intervention’: “Before the mass screening intervention (1950-1956), tuberculosis notification rates were declining at 2.3% per year, and rates doubled in the year of the intervention (1957). Post-intervention, tuberculosis notification rates declined at 5.4% per year, and there were an estimated 4,599 pulmonary notifications averted.”

2. Please ensure that the study is reported according to the STROBE (or appropriate STOBE extension) guideline (available from: https://www.equator-network.org/reporting-guidelines/strobe) and include the completed STROBE (or STROBE extension) checklist as Supporting Information. Please add the following statement, or similar, to the Methods: "This study is reported as per the Strengthening the Reporting of Observational Studies in Epidemiology (STROBE) guideline (S1 Checklist)." When completing the checklist, please use section and paragraph numbers, rather than page numbers. 

3. To help us extend the reach of your research, please provide any X (formerly known as Twitter) handle(s) that would be appropriate to tag, including your own, your co-authors’, your institution, funder, or lab. Please enter in the submission form any handles you wish to be included when we post about this paper.

Comments from Reviewers:

Reviewer #3: 

The authors have addressed my points. Our remaining differences are very minor and I would not want to push them.

Michael Dewey

[LINK]

---

## [Editor Report · Decision Letter 3]

3 Oct 2024

Dear Dr MacPherson, 

On behalf of my colleagues and the Academic Editor, Amitabh Bipin Suthar, I am pleased to inform you that we have agreed to publish your manuscript "Impact of active case finding for tuberculosis with mass chest X-ray screening in Glasgow, Scotland, 1950-1963: an epidemiological analysis of historical data" (PMEDICINE-D-24-02288R3) in PLOS Medicine.

I appreciate your thorough responses to the final editorial comments. We look forward to publishing your manuscript, and there is only one small point to address prior to publication. We ask that you please add line numbers to your STROBE checklist and submit an updated version. If you have any questions or concerns at any stage of the post-accept process, please feel free to contact me at hvanepps@plos.org.

PRESS

Kind regards,

Heather

Heather Van Epps, PhD 

Executive Editor 

PLOS Medicine